# LoMOE: Localized Multi-Object Editing via Diffusion

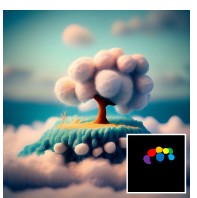 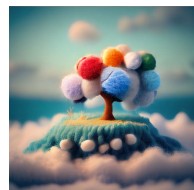 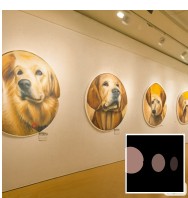 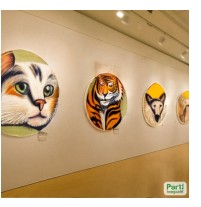 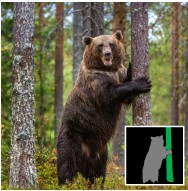 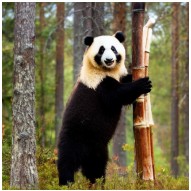

"A tree on top of a cloud covered island"

{V,I,B,G,Y,O,R} cottony

"A row of paintings of dogs on a wall"

cat  tiger  wolf

"A brown bear standing against a tree trunk"

panda  bamboo tree

**Figure 1: Representative results of LoMOE on diverse images: Our algorithm can handle *multi-object* edits in *one go*. The first image in each example depicts the original image with the input mask (can be obtained using bounding boxes). Below each image is the text caption describing the image and the text prompts (in *color*) describing the edits. The second image depicts the edited image using LoMOE. Observe, that our method handles intricate localized object details such as multiple-cloud coloring, editing animals on a wall painting, and lastly, editing tree and animal classes.**

## ABSTRACT

Recent developments in diffusion models have demonstrated an exceptional capacity to generate high-quality prompt-conditioned image edits. Nevertheless, previous approaches have primarily relied on textual prompts for image editing, which tend to be less effective when making precise edits to specific objects or fine-grained regions within a scene containing single/multiple objects. We introduce a novel framework for zero-shot localized multi-object editing through a multi-diffusion process to overcome this challenge. This framework empowers users to perform various operations on objects within an image, such as adding, replacing, or editing **many** objects in a complex scene **in one pass**. Our approach leverages foreground masks and corresponding simple text prompts that exert localized influences on the target regions resulting in high-fidelity image editing. A combination of cross-attention and background preservation losses within the latent space ensures that the characteristics of the object being edited are preserved while simultaneously achieving a high-quality, seamless reconstruction of the background with fewer artifacts compared to the state-of-the-art (SOTA). We also curate and release a dataset dedicated to multi-object editing, named LoMOE-Bench. Our experiments against existing SOTA demonstrate the improved effectiveness of our approach in terms of both image editing quality, and inference speed.

## CCS CONCEPTS

• **Computing methodologies → Image processing**.

## KEYWORDS

Image Editing, Generative Modelling, Diffusion Models

## 1 INTRODUCTION

Diffusion models [39–41] have exhibited an outstanding ability to generate highly realistic images based on text prompts. However, text-based editing of multiple fine-grained objects precisely at given locations within an image is a challenging task. This challenge primarily stems from the inherent complexity of controlling diffusion models to specify the accurate spatial attributes of an image, such as the scale and occlusion during synthesis. Existing methods for textual image editing use a global prompt for editing images, making it difficult to edit in a specific region while leaving other regions unaffected [6, 32]. Thus, this is an important problem to tackle, as real-life images often have multiple subjects and it is desirable to edit each subject independent of other subjects and the background while still retaining coherence in the composition of the image. To this end, we propose **Lo**calized **M**ulti-**O**bject **E**diting (LoMOE).

Our method draws inspiration from the recent literature on compositional generative models [3, 18, 25]. It inherits generality without requiring training, making it a zero-shot solution similar to [3]. We utilize a pre-trained StableDiffusion 2.0 [40] as our base generative model. Our approach involves the manipulation of the diffusion trajectory within specific regions of an image earmarked for editing. We employ prompts that exert a localized influence on these regions while simultaneously incorporating a global prompt to guide the overall image reconstruction process that ensures a coherent composition of foreground and background with minimal/imperceptible artifacts. To initiate our editing procedure, we employ the inversion of the original image as a starting point, as proposed in [37]. For achieving high-fidelity, human-like edits in

our images, we employ two crucial steps: **(a)** cross-attention matching and **(b)** background preservation. These preserve the integrity of the edited image by guaranteeing that the edits are realistic and aligned with the original image. This, in turn, enhances the overall quality and perceptual authenticity of the final output. Additionally, we also curate a novel benchmark dataset, named LoMOE-Bench for multi-object editing. Our contributions in this paper are as follows:

(1) We present a framework LoMOE, for zero-shot text-based localized multi-object editing based on Multi-diffusion [3]. Our framework facilitates multiple edits in a single iteration via enforcement of cross-attention and background preservation, resulting in high fidelity and coherent image generation.

(2) We introduce a new benchmark dataset for evaluating the multi-object editing performance of existing frameworks, termed LoMOE-Bench.

## 2 RELATED WORK

**Image Synthesis and Textual Guidance:** Text-to-image synthesis has made significant strides in recent years, with its early developments rooted in RNNs [31] and GANs [17], which were effective in generating simple objects such as flowers, dogs and cats but struggled in generating complex scenes, especially with multiple objects [4]. These models have now been superseded by diffusion-based methods which produce photorealistic images, causing a paradigm shift [21, 40, 41]. In a separate line of work, CLIP [38] was introduced, which is a vision-language model trained on a dataset of 400 million image-text pairs using techniques such as contrastive training. The rich embedding space CLIP provides has enabled various multi-modal applications such as text-based imaged generation [12, 13, 16, 24, 36, 39, 40, 46].

**Compositional Diffusion Model:** Kim *et al.* [25] observe text-to-image models fail to adhere to the positional/layout prompting via text. Thus, compositional diffusion models try to address the task of image generation conditioned on masks, where each mask is associated with a text prompt. In Make-a-Scene [15], the initial step involves predicting a segmentation mask based on the provided text. Subsequently, this generated mask is employed in conjunction with the text to produce the final predicted image. Methods such as ControlNet and GLIGEN [29, 49] have propose fine-tuning for synthesizing images given text descriptions and spatial controls based on adapters. Finally, methods like [3, 18, 25], aim to utilise the pre-trained models and masked regions with independent prompts to generate images without re-training.

**Image Editing:** Paint-by-Word [1] was one of the first approaches to tackle the challenge of zero-shot local text-guided image manipulation. But this method exclusively worked with generated images as input and it required a distinct generative model for each input domain. Later, Meng *et al.*[32] showed how the forward diffusion process allows image editing by finding a common starting point for the original and the editing image. This popularised inversion among image editing frameworks such as [24, 37]. This approach was further improved upon by adding a structure prior to the editing process using cross-attention matching [19, 37]. Moreover, there have been improvements in inversion techniques producing higher quality reconstruction which results in more faithful edits [23, 33]. However, many of the aforementioned methods generate the whole

image from the inversion. This compromises the quality of reconstruction in regions where the image was not supposed to be edited.

Recently [5, 8] try to address the problem with the above mask-free methods by incorporating an implicit masking strategy based on cross-attention masks similar to [11]. Thus reinforcing the notion that masking (either implicit or explicit) is essential for restricting the generation process to a certain region [2, 34]. However, when it comes to multi-object editing, these methods fall short on 3 counts: (1) editing multiple regions in one pass, (2) maintaining consistency between the edited and the non-edited regions of the image, (3) accumulating error over the multiple edit passes. Our method explicitly takes care of these aspects of image editing while incorporating all the advancements of our predecessor methods.

## 3 PROPOSED METHOD

**Problem Statement:** In a multi-object editing scenario, the objective is to simultaneously make local edits to several objects within an image. Formally, we are given a pretrained diffusion model $\Phi$, an image $\mathbf{x}_0$ from image space $\mathcal{X} \subset \mathbb{R}^{w \times h \times 3}$ ($\mathbf{x}_0 \in \mathcal{X}$ (for stable diffusion-based models, $\mathcal{X} \subset \mathbb{R}^{512 \times 512 \times 3}$), and $N$ binary masks $\{M_1, \cdots, M_N\}$ along with a corresponding set of prompts $\{c_1, \cdots, c_N\}$, where $c_i \in C$, the space of encoded text prompts. They are used to obtain an edited image $\mathbf{x}_*$ such that the editing process precisely manifests at the locations dictated by the masks, in accordance with the guidance provided by the prompts.

**Overview of LoMOE:** **Lo**calized **M**ulti-**O**bject Image **E**diting (LoMOE) comprises of three key steps **(a)** Inversion of the original image $\mathbf{x}_0$ to obtain the latent code $x_{inv}$, which initiates the editing procedure and ensures a coherent and controlled edit **(b)** Applying the MultiDiffusion process for localized multi-object editing to limit the edits to mask-specific regions, and **(c)** Attribute and Background Preservation via cross attention and latent background preservation to retain structural consistency with the original image. Fig. 2 depicts an overview of our method.

### 3.1 Inversion for Editing

In this work, we employ a pretrained Stable Diffusion [40] model, denoted as $\Phi$. This model encodes an input image $\mathbf{x}_0 \in \mathbb{R}^{512 \times 512 \times 3}$ into a latent code $x_0 \in \mathcal{E} \subset \mathbb{R}^{64 \times 64 \times 4}$.

Given an image $\mathbf{x}_0$ and it's corresponding latent code $x_0$, *inversion* entails finding a latent $x_{inv}$ which reconstructs $x_0$ upon sampling. We adopt a deterministic DDIM reverse process to model the *inversion* step [37]. This process is deterministic when $\sigma_t = 0$ $\forall\, t \in [T]$, where $\sigma \in \mathbb{R}_+^T$ parameterizes the family $Q$ of inference distributions [41] and $T$ is the number of timesteps. The latent $x_{inv} = x_T$ and the intermediate latents are related by

$$x_{t+1} = \sqrt{\alpha_{t+1}} \left( \frac{x_t - \sqrt{1 - \alpha_t}\, \epsilon_\theta\, (x_t, t)}{\sqrt{\alpha_t}} \right) + \sqrt{1 - \alpha_{t+1}}\, \epsilon_\theta\, (x_t, t) \quad (1)$$

where $\alpha_t$ represents a prefixed noise schedule and $\epsilon_\theta(x_t, t)$ is a neural network trained to predict the noise $\epsilon_t$ added to a sample $x_t$. This network can also be conditioned on text, images, or embeddings [22], denoted by $\epsilon_\theta(x_t, t, c, \oslash)$, where $c$ is the encoded condition and $\oslash$ is the null condition. In LoMOE, $\epsilon_\theta$ is conditioned

**Figure 2: LoMOE comprises of 3 main steps:** Inversion **(Sec. 3.1) produces $x_{inv}$ and $c_0$ corresponding to input $\mathbf{x}_0$. A** MultiDiffusion **process (Sec. 3.2) helps restrict the edits to regions $M_1, M_2$ guided by $c_1, c_2$. The** Preservation of Attributes **(Sec. 3.3) is achieved via $\mathcal{L}_{xa}$ and $\mathcal{L}_b$, using reference cross-attention maps and background latents obtained through a** reconstruction **process.**

on $c_0$, a text prompt encoded using CLIP [38], during *inversion*. The underlying prompt is generated utilizing a text-embedding framework such as BLIP [28] on the image $\mathbf{x}_0$.

Additionally, at each step during the *inversion* process, we softly enforce gaussianity using a pairwise regularization $\mathcal{L}_{pair}$ [37] and a divergence loss $\mathcal{L}_{KL}$ [26] weighted by $\lambda$. This adaptation is inspired by findings in [37], which highlighted deviations from the desired statistical properties of uncorrelated, white gaussian noise in the noise maps generated by $\epsilon_\theta$, leading to poor editability. Details of these losses can be found in Sec. 1 of the supplementary.

The *inversion* step offers a solid foundation for the editing process, outperforming random latent initialization (Ref. Supplementary Sec. 2.1). However, employing a standard diffusion process for editing poses limitations in controlling local regions within the image via simple prompts. To address this challenge, we adopt a MultiDiffusion approach [3] for localized multi-object editing.

## 3.2 Diffusion for Multi-Object Editing

For a diffusion model $\Phi$, the backward process entails generating a sequence of latents $\{x_i\}_{i=T-1}^0$ starting from $x_T$, progressively denoising it over time. Here, $x_{t-1} = \Phi(x_t|c)$, where $c$ is the encoded condition. Utilizing a deterministic DDIM reverse process,

$$x_{t-1} = \sqrt{\alpha_{t-1}}\left(\frac{x_t - \sqrt{1-\alpha_t}\,\epsilon_\theta\,(x_t, t, c, \oslash)}{\sqrt{\alpha_t}}\right) + \sqrt{1-\alpha_{t-1}}\,\epsilon_\theta\,(x_t, t, c, \oslash) \quad (2)$$

By running this backward process with $x_T = x_{inv}$ and the source prompt $c_0$, we obtain a reconstructed version, $x_0'$, of the original latent code $x_0$. This step is termed the *reconstruction* phase. To address any deviations between $x_0'$ and $x_0$, we adopt a strategy of preserving noise latents during the inversion process [23]. Additionally, we store the latents $x_t'$ and cross-attention maps $\bar{A}_t^r$ (Sec. 3.3.1) at each timestep $t$.

A simple approach to edit $\mathbf{x}_0$ involves running a backward process with $x_T = x_{inv}$ and guiding it using a target prompt [32]. However, this method applies prompt guidance across the entire image,

rendering the output susceptible to unintended edits. Thus, we propose a localized prompting solution, confining edits to a masked region. To edit $N$ regions corresponding to $N$ masks concurrently, one might initially consider utilizing $N + 1$ distinct diffusion processes $\{\Phi(x_t^j|c_j)\}_{j=0}^N$. Here, $\{x_t^j, c_j\}_{j\geq 1}$ denote the latent code and encoded prompt for mask $j$, while $\{x_t^0, c_0\}$ correspond to those of the background (source image $\mathbf{x}_0$). However, LoMOE adopts a single MultiDiffusion process [3] denoted by $\Psi$ for zero-shot conditional editing of regions within all the masks.

Given masks $\{M_1, \cdots, M_N\}$ and $M_0 = 1 - min\{\bigcup_{i=1}^N M_i, 1\}$, with a corresponding set of encoded text prompts $z = (c_0, c_1, \cdots, c_N)$, the goal is to come up with a mapping function $\Psi : \mathcal{E} \times C^{N+1} \to \mathcal{E}$, solving the following optimization problem:

$$\Psi(y_t, z) = \underset{y_{t-1}}{\text{argmin}}\ \mathcal{L}_{md}(y_{t-1}|y_t, z) \quad (3)$$

Starting from $y_T$, $\Psi$ generates a sequence of latents $\{y_i\}_{i=T-1}^0$ during the backward process, where $y_{t-1} = \Psi(y_t|z)$. The objective in Eq. 3 is designed to follow the denoising steps of $\Phi$ as closely as possible, enforced using the constraint $\mathcal{L}_{md}$ defined as:

$$\mathcal{L}_{md}(y_{t-1}|y_t, z) = \sum_{i=0}^N \left\| M_i \otimes \left[y_{t-1} - \Phi(x_t^i \mid c_i)\right] \right\|^2 \quad (4)$$

where $\otimes$ is the Hadamard product. The optimization problem in Eq. 3 has a closed-form solution given by:

$$\Psi(y_t, z) = \sum_{i=0}^N \frac{M_i}{\sum_{j=0}^N M_j} \otimes \Phi\left(x_t^i \mid c_i\right) \quad (5)$$

Thus, editing in LoMOE is accomplished by running a backward process using $\Psi$ with $x_T^0 = x_T^1 = \cdots = x_T^N = x_{inv}$ and in turn $y_T = x_{inv}$ via a deterministic DDIM reverse process for $\Phi$ (i.e, $\Phi\left(x_t^i \mid c_i\right)$ is given by Eq. 2). This step is termed the *edit* phase. Additionally, the latents and attention maps stored during the *reconstruction* phase are used to define losses (Sec. 3.3) that guide the *edit*.

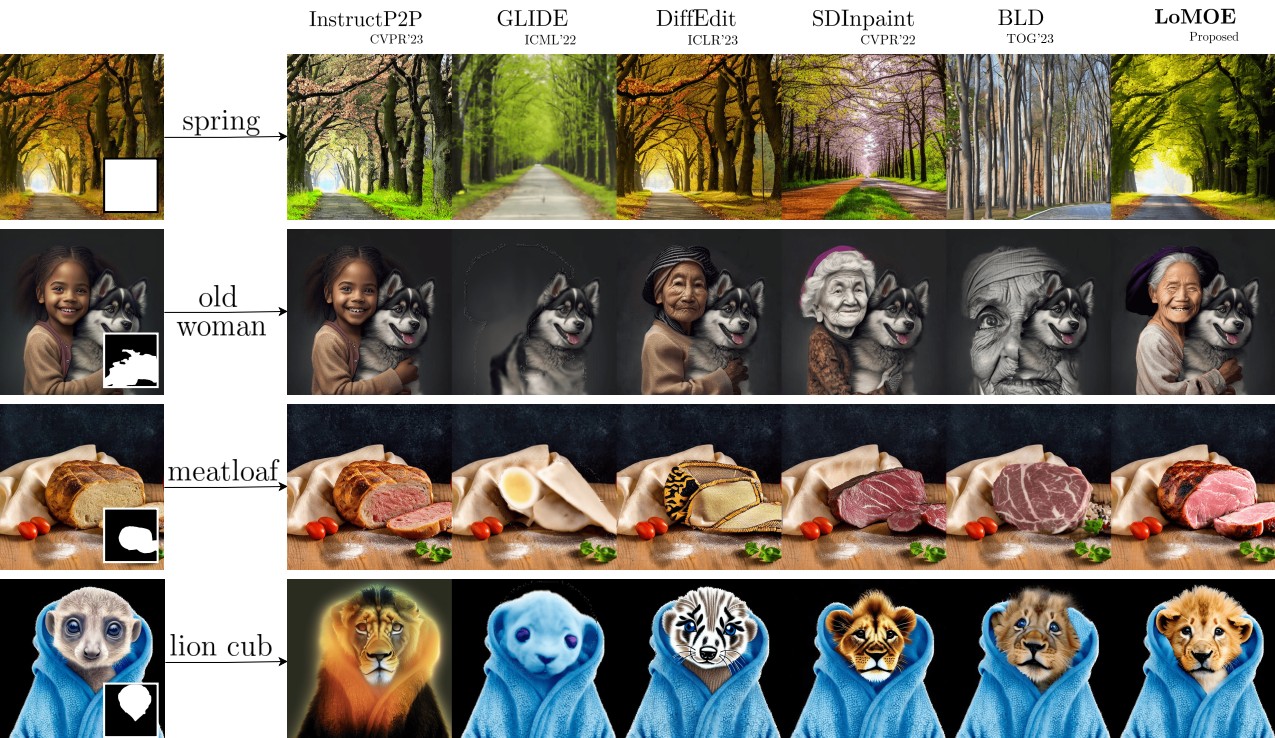

**Figure 3: Comparison among contemporary methods for Single Object Edits:** We observe that InstructP2P [6] tends to modify the whole image. GLIDE [35] often removes the subject of the edit in cases where it fails to generate the edit. DiffEdit [11] often fails to make a successful edit although it is based on Stable Diffusion. BLD [2] and SDInpaint [40] don't preserve the structure of the input and make unintended attribute edits to the masked subject. Finally, we observe that our proposed LoMOE makes the intended edit, preserves the unmasked region and avoids unintended attribute edits.

*3.2.1* **Bootstrapping.** To enable $\Psi(y_t|c_i)$ to focus on region $M_i$ during the early stages of the backward process (up to timestep $T_b$, referred to as the bootstrap parameter), while incorporating the entire image context later on [3], we introduce a time-dependency in $y_t$, as follows:

$$y_t = \begin{cases} M_i \cdot y_t + (1 - M_i) \cdot b_t, & \text{if } t < T_b \\ y_t, & \text{otherwise} \end{cases} \quad (6)$$

where $b_t$ serves as a background and is obtained by noising the encoded version of a random image with a constant color to the noise level of timestep $t$, i.e. $b_t = \xi(\mathbf{x})$ where $\mathbf{x} \in \mathcal{X}$ and $\xi$ is the Stable Diffusion encoder. This contributes to improved fidelity in generated images, particularly in scenarios involving tight masks.

## 3.3 Attribute Preservation during Editing

While $\Psi$ addresses multi-object editing, it faces challenges in (1) maintaining structural consistency with the source image and (2) faithfully reconstructing the background. To address these shortcomings, we introduce losses $\mathcal{L}_{xa}$ and $\mathcal{L}_b$ as post-hoc guidances. These losses are jointly optimized at each iteration during the *edit* process, thereby constraining the diffusion process.

*3.3.1* **Cross-Attention Preservation.** Diffusion models such as Stable Diffusion [40] incorporate cross-attention (CA) layers [43]

within $\epsilon_\theta$ to effectively condition their generation on text. These layers facilitate interaction between image and text modalities during denoising, resulting in spatial attention maps for each textual token. These attention maps are represented as:

$$\bar{A} = \text{Softmax}\left(\frac{QK^T}{\sqrt{d}}\right) \quad (7)$$

where $Q$ denotes the projection of intermediate spatial features from $\epsilon_\theta$ onto a query matrix $W_Q$, $K$ denotes the projection of the text embedding $c$ onto a key matrix $W_K$, $d$ signifies the latent projection dimension, and $\bar{A}_{i,j}$ represents the weight of the $j^{th}$ text token on the $i^{th}$ pixel.

Studies [19, 42] validate that UNet encodings, especially CA maps, encode valuable information about structure and spatial layout. Consequently, constraints on intermediate CA maps can guide sampling and control generation, as shown in [14, 37]. While techniques such as mask-based blending [11] and attention injection [19] aid in preserving structure, they often yield suboptimal results (Ref. Table 1). Additionally, [19] suggests that attention injection may overly constrain geometry, favoring a softer constraint.

In LoMOE, we employ a soft CA guidance through $\mathcal{L}_{xa}$, controlled by $\lambda_{xa}$. During the *edit* process, we update the attention maps $(\bar{A}_t^e)$

| Method | | Mask | Target CLIP Score (↑) | Background LPIPS (↓) | Structural Distance (↓) | IR (↑) | HPS (↑) | Source CLIP Score (↑) | Background SSIM (↑) |
|---|---|---|---|---|---|---|---|---|---|
| Input | | - | $23.584 \pm 0.221$ | - | - | - | - | $25.639 \pm 0.178$ | - |
| SDEdit | [32] | ✗ | $23.042 \pm 0.250$ | $0.199 \pm 0.0071$ | $0.084 \pm 0.0035$ | $-0.600 \pm 0.074$ | $0.237 \pm 0.003$ | $21.362 \pm 0.266$ | $0.788 \pm 0.0086$ |
| I-P2P | [6] | ✗ | $25.038 \pm 0.216$ | $0.242 \pm 0.0123$ | $0.090 \pm 0.0042$ | $-0.217 \pm 0.079$ | $0.254 \pm 0.003$ | $22.513 \pm 0.273$ | $0.762 \pm 0.0105$ |
| NTI (w/P2P) | [33] | ✗ | $25.152 \pm 0.226$ | $0.098 \pm 0.0069$ | $0.074 \pm 0.0039$ | $0.205 \pm 0.073$ | $0.257 \pm 0.003$ | $23.415 \pm 0.247$ | $0.842 \pm 0.0082$ |
| MasaCtrl | [7] | ✗ | $24.389 \pm 0.227$ | $0.197 \pm 0.0074$ | $0.085 \pm 0.0037$ | $-0.465 \pm 0.073$ | $0.238 \pm 0.003$ | $24.034 \pm 0.231$ | $0.782 \pm 0.0087$ |
| GLIDE | [34] | ✓ | $24.299 \pm 0.215$ | $0.104 \pm 0.0041$ | $0.094 \pm 0.0035$ | $-0.646 \pm 0.068$ | $0.215 \pm 0.003$ | $22.756 \pm 0.235$ | $0.938 \pm 0.0031$ |
| DiffEdit | [11] | ✓ | $24.094 \pm 0.234$ | $0.057 \pm 0.0019$ | $0.076 \pm 0.0036$ | $-0.381 \pm 0.074$ | $0.247 \pm 0.003$ | $23.269 \pm 0.248$ | $0.875 \pm 0.0063$ |
| SDInpaint | [40] | ✓ | $25.556 \pm 0.230$ | $0.067 \pm 0.0072$ | $0.093 \pm 0.0057$ | $0.149 \pm 0.077$ | $0.253 \pm 0.002$ | $23.068 \pm 0.246$ | $0.854 \pm 0.0095$ |
| BLD | [2] | ✓ | $25.867 \pm 0.206$ | $0.058 \pm 0.0021$ | $0.077 \pm 0.0034$ | $0.374 \pm 0.069$ | $0.263 \pm 0.002$ | $22.761 \pm 0.238$ | $0.877 \pm 0.0062$ |
| LoMOE | | ✓ | $26.074 \pm 0.201$ | $0.054 \pm 0.0022$ | $0.066 \pm 0.0031$ | $0.457 \pm 0.069$ | $0.271 \pm 0.002$ | $23.545 \pm 0.219$ | $0.885 \pm 0.0060$ |

Table 1: Comparison with different baselines for Single-Object Edits: We use a large array of *classical* and *neural* metrics that provide valuable statistical insights regarding the edit properties of considered methods. The best and the second best methods are highlighted. In particular, LoMOE outperforms the baselines on all *neural* metrics indicating realistic image generation. Additionally, LoMOE also performs faithful edits, as indicated by it's high *classical* metrics.

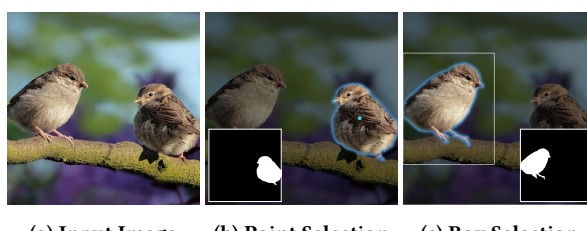

(a) Input Image    (b) Point Selection    (c) Box Selection

Figure 4: Mask Generation using SAM [27].

to match those during *reconstruction* $(\bar{A}^r_t)$ at each timestep $t$ by

$$\bar{A}^e_t \leftarrow \epsilon_\theta \left( [x^0_t, \cdots, x^N_t], t, [c_0, \cdots, c_N], \oslash \right) \qquad (8)$$

$$\mathcal{L}_{xa} = \|\bar{A}^r_t - \bar{A}^e_t\|_2 \qquad (9)$$

Additionally, we incorporate a temperature parameter $\tau$ in Eq. 7 to ensure distributional smoothness (Ref. Supplementary Sec. 2.1).

### 3.3.2 *Background Preservation*.
In order to preserve the *background* in the output, we match the backgrounds of the latents during the *edit* process $(y^*_t)$ with those stored during *reconstruction* $(x'_t)$ at each timestep using a loss $\mathcal{L}_b$.

$$\mathcal{L}_b = \|M_0 \cdot (y^*_t - x'_t)\|_2 \qquad (10)$$

where $M_0$ is the *background* mask.

This approach is preferred over simple copy-pasting of the background [11] to ensure natural and photorealistic edits while avoiding border artifacts through improved blending of multiple regions generated by separate diffusion processes.

### 3.3.3 *Joint Optimization*.
During each timestep of the *edit* process, we update the attention maps and latent vectors by optimizing the combined loss:

$$\Delta x^i_t = \nabla_{x^i_t} \left( \lambda_{xa} \mathcal{L}_{xa} + \lambda_b \mathcal{L}_b \right) \ \forall i \in [0, N] \qquad (11)$$

$$\left[ x^0_t, \cdots, x^N_t \right] = \left[ x^0_t - \Delta x^0_t, \cdots, x^N_t - \Delta x^N_t \right] \qquad (12)$$

where $\lambda_{xa}$ and $\lambda_b$ represent the weights assigned to the cross-attention and background preservation losses, respectively. The updated latent is given by:

$$y^*_{t-1} = \sum_{i=0}^{N} \frac{M_i}{\sum_{j=0}^{N} M_j} \otimes \Phi \left( x^i_t \mid c_i \right) \qquad (13)$$

where $\Phi$ represents the diffusion model, $\{M_1, \cdots, M_N\}$ are foreground masks, and $M_0$ is the background mask with corresponding encoded prompts $\{c_1, \cdots, c_N\}$ and $c_0$ respectively. Additionally, $x^i_t$ is the latent associated with mask $M_i$ at timestep $t$.

### 3.4 Implementation Details
We utilized Stable Diffusion v2.0 as our pretrained model $\Phi$. Additionally, we set the hyperparameters: $\lambda_b = 1.75, \lambda_{xa} = 1.00, \tau = 1.25$, and $T_b = 10$, based on empirical validation conducted on a held-out set comprising ten images. The majority of our experiments were conducted on a system equipped with a GeForce RTX-3090 with 24 GB of memory. For multi-object edits involving more than five masks, we utilized an A6000 GPU with 48 GB of memory. The code will be made available post-acceptance.

## 4 EXPERIMENTAL SETTING
We consider two sets of experiments: (a) single-object edits and (b) multi-object edits. For the multi-object editing experiments, while LoMOE can be employed as it is, we resort to iterative editing for other methods, specifically dealing with mask-based methods from Table 1. We report both qualitative and quantitative outcomes of our experiments.

### 4.1 Datasets
For single-object edits, we utilized a modified subset of the PIE-Bench [23] dataset, supplemented with images from AFHQ [9], COCO [30], and Imagen [44]. For multi-object edits, we introduce a new dataset named LoMOE-Bench, comprising ~1000 edit operations featuring images with 2 to 7 masks, each paired with corresponding text prompts. Each image has 4 masks on average yielding 15 edit combinations per image through combinatorial selection ($\sum_{i=1}^{4} {}^4C_i$),

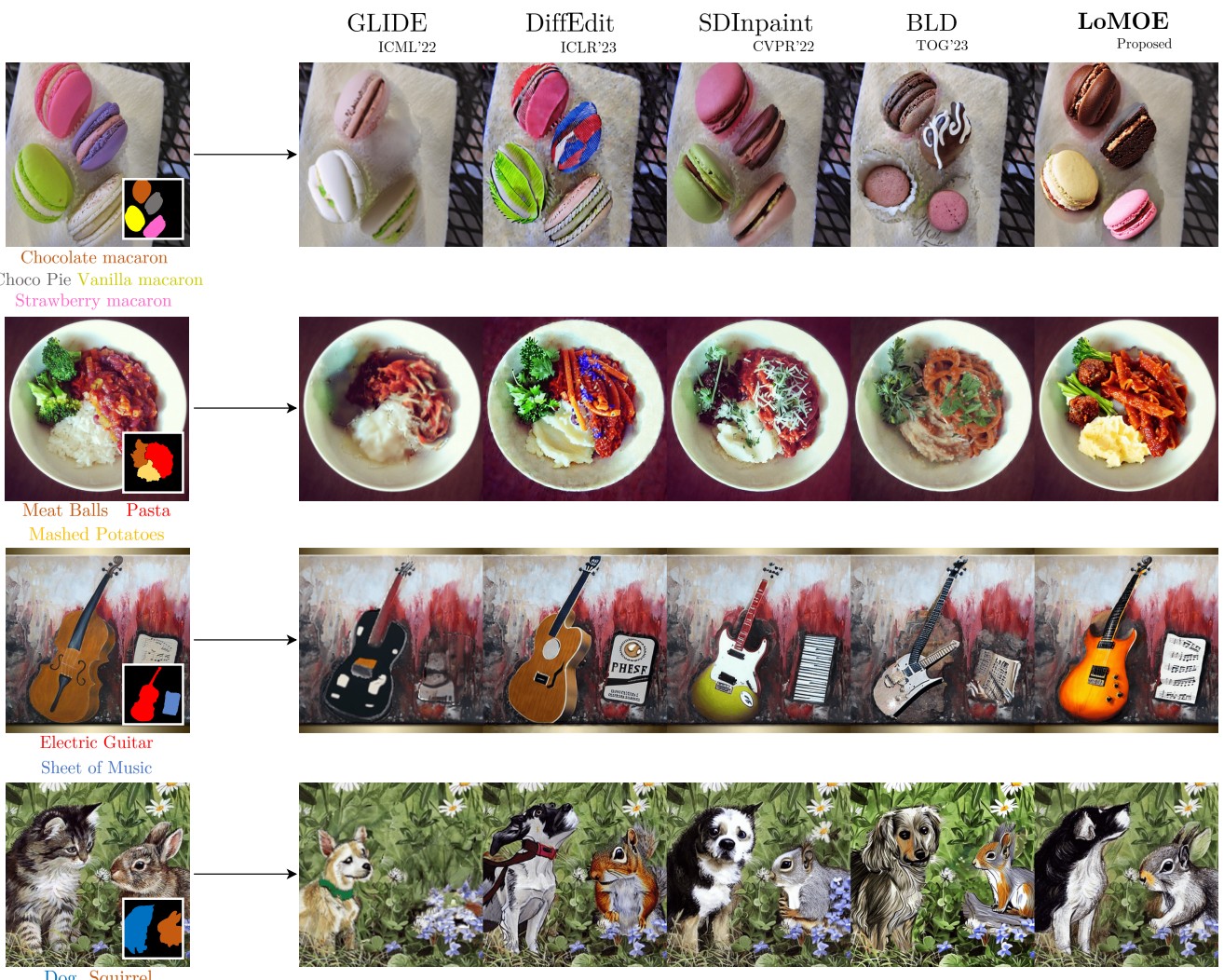

**Figure 5: Comparison with contemporary methods for Multi-Object Edits: While the baselines are either unable to make the edit, accumulate artifacts, edit the unmasked region, or make unintended attribute edits, LoMOE is able to faithfully edit in accordance with the target prompts.**

resulting in ~1,000 operations across diverse images. The details of the curated dataset can be found in Sec. 4.1 of the supplementary material. The LoMOE-Bench dataset will be made public in due time.

To obtain masks for LoMOE-Bench, we employ SAM [27], where users can generate masks either by clicking on objects of interest or by drawing bounding boxes around them, as illustrated in Fig. 4. Additionally, for masks required in *addition* tasks for both datasets, we developed a simple Python GUI where users can draw masks directly onto the target regions of the images.

## 4.2 Baseline Methods

We benchmark LoMOE against SOTA, including SDEdit [32], Instruct-Pix2Pix (I-P2P) [6], MasaCtrl [7], Null Text Inversion with Prompt-to-Prompt (NTI w/ P2P) [33], GLIDE [34], DiffEdit [11], Stable Diffusion Inpaint (SDInpaint) [40] and Blended Latent Diffusion

(BLD) [2]. Official implementations were used for all methods, except for SDEdit and DiffEdit. GLIDE, DiffEdit, SDInpaint, BLD, and LoMOE leverage masks, whereas the other methods operate on the whole image. Additionally, there are differences among the methods in terms of the types of text prompts they require. SDEdit, DiffEdit, NTI (w/P2P) and MasaCtrl necessitate both source and target text prompts, and I-P2P takes edit instructions as prompts, prompting us to extend PIE-Bench to accommodate these methods. Similar to LoMOE, GLIDE, SDInpaint, and BLD only use edit prompts corresponding to the masks. Finally, given the considerably noisy masks generated by DiffEdit, we opted to provide it with ground truth masks.

| Method | Single Pass | Target CLIP Score (↑) | Background LPIPS (↓) | Structural Distance (↓) | IR (↑) | HPS (↑) | Source CLIP Score (↑) | Background SSIM (↑) |
|---|---|---|---|---|---|---|---|---|
| Input | - | 22.489 ± 0.236 | - | - | - | - | 26.956 ± 0.141 | - |
| GLIDE [34] | ✗ | 22.754 ± 0.526 | 0.192 ± 0.0151 | 0.085 ± 0.0065 | -1.224 ± 0.052 | 0.187 ± 0.002 | 27.038 ± 0.308 | 0.894 ± 0.0104 |
| DiffEdit [11] | ✗ | 23.898 ± 0.445 | 0.188 ± 0.0119 | 0.071 ± 0.0063 | -0.574 ± 0.069 | 0.227 ± 0.002 | 26.417 ± 0.306 | 0.756 ± 0.0168 |
| SDInpaint [40] | ✗ | 24.804 ± 0.457 | 0.302 ± 0.0155 | 0.089 ± 0.0129 | -0.214 ± 0.063 | 0.244 ± 0.002 | 26.506 ± 0.302 | 0.761 ± 0.0204 |
| BLD [2] | ✗ | 25.394 ± 0.450 | 0.126 ± 0.0086 | 0.074 ± 0.0062 | 0.043 ± 0.070 | 0.242 ± 0.002 | 26.330 ± 0.268 | 0.800 ± 0.0150 |
| LoMOE | ✓ | 26.154 ± 0.187 | 0.107 ± 0.0040 | 0.066 ± 0.0027 | 0.527 ± 0.061 | 0.264 ± 0.002 | 25.959 ± 0.111 | 0.826 ± 0.0073 |

**Table 2: Comparison with SOTA for Multi-Object Edits: We use a large array of *classical, neural* and *aesthetic* metrics that provide valuable statistical insights regarding the edit properties of considered methods. The best and the second best have been highlighted. We observe that only LoMOE has a higher target CS compared to source CS.**

## 4.3 Metrics

We quantitatively analyze the edited images on a set of *neural* metrics, namely Clip Score (CS) [20] with both source and target prompts, Background (BG)-LPIPS [50], and Structural Distance [10]. Additionally, we employed *classical* metrics such as BG-SSIM [45]. The *neural* metrics evaluate the perceptual similarity of the image, emphasizing realism. On the other hand, *classical* metrics focus on pixel-level similarity and doesn't comment on the realism or quality of the edit. In contrast to previous approaches, we propose evaluating edits based on the **target CS** and offer target prompts for all images in both datasets. This approach enhances the effectiveness of measuring edit quality, as a high target CS indicates successful editing. Finally, we also use state-of-the-art image *aesthetic* metrics such as Image Reward (IR) [48] and Human Preference Score (HPS) [47] which have not been used previously to evaluate editing, to the best of our knowledge. These metrics validate which method produces images that are pleasing to the human eye. To ensure robustness in our assessments, we averaged all the metrics over 5 seeds and reported the average standard error for all methods. Additionally, we conduct a subjective evaluation experiment to assess the quality of edits, described in Sec. 5.4.

## 5 RESULTS AND DISCUSSION

### 5.1 Single Object Edits

In comparing LoMOE with various baselines Table 1, LoMOE demonstrates superior *neural* metrics, highlighting its proficiency in maintaining fidelity with source image and target prompt while making realistic edits. However, GLIDE outperforms LoMOE in *classical* BG-SSIM, suggesting a trade-off between realism and pixel-wise faithfulness, as observed in prior works [32]. While GLIDE does well on BG-SSIM due to its inpainting model design, it falls short on *neural* and *aesthetic* metrics, resulting in less realistic/incorrect edits. Other methods like MasaCtrl, NTI (w/P2P), and I-P2P perform well on target CS, but lack in other aspects, especially *background* metrics, due to their operation without a mask. Notably, instances where the target CS is close to the first-row in Table 1 suggest the absence of applied edits. Therefore, target CS is the most important metric in this context. Masked methods like DiffEdit, BLD, and SD-Inpaint collectively rank second best across most metrics, indicating *the preference for utilizing a mask in our edit context.* Qualitative

evaluation in Fig. 3 provides visual comparisons. Finally, LoMOE achieves the highest scores in the *aesthetic* metrics indicating that it's edits have the least artifacts and are most pleasant to humans. Further, Fig. 6 validates this in the user study.

### 5.2 Multi-Object Edits

Similar to our observations in single-object editing, LoMOE exhibits superior performance across all *neural* and *aesthetic* metrics in multi-object editing, except for source CS. This deviation is anticipated, given the substantial image transformations in multi-object editing. Ideally, such transformations lead to images that are markedly different from the source prompt and more aligned with the target prompt. Therefore, elevated BG-LPIPS and Structural Distance better indicate perceptual quality, while a high target CS signifies successful editing. Conversely, all other methods display a considerably lower target CS compared to source CS, indicating unsuccessful edits. Intuitively, as the number of edited objects increases, the source CS tends to decrease, while the target CS tends to increase. Furthermore, given our single-pass approach, we achieve significant savings in edit time compared to methods that perform multi-edits iteratively. Additional details can be found in Sec. 3.4 of the supplementary. Fig. 5 shows qualitative results on all the compared methods on a few sample images. This demonstrates

| $\mathcal{L}_{xa}$ | $\mathcal{L}_b$ | Source CLIP Score (↑) | Structural Distance (↓) | Target CLIP Score (↑) |
|---|---|---|---|---|
| ✗ | ✗ | 23.0906 | 0.0763 | 26.2555 |
| ✗ | ✓ | 23.3925 | 0.0728 | **26.2662** |
| ✓ | ✗ | **23.6611** | 0.0699 | 26.1338 |
| ✓ | ✓ | 23.5445 | **0.0661** | 26.0740 |

| $\mathcal{L}_{xa}$ | $\mathcal{L}_b$ | Background LPIPS (↓) | Background PSNR (↑) | Background SSIM (↑) |
|---|---|---|---|---|
| ✗ | ✗ | 0.1088 | 26.4474 | 0.8537 |
| ✗ | ✓ | 0.0554 | 30.1475 | 0.8818 |
| ✓ | ✗ | 0.0749 | 26.9587 | 0.8698 |
| ✓ | ✓ | **0.0546** | **30.3154** | **0.8847** |

**Table 3: Ablation Study: We observe that both our losses complement each other and result in improved metrics**

LoMOE's impressive performance in preserving the intricate details during edits.

## 5.3 Ablation Studies

To assess the significance of each loss component in LoMOE, we conducted a comprehensive ablation study, maintaining a fixed seed, $\tau$ and $T_b$. The findings presented in Table 3 reveal that incorporating $\mathcal{L}_{xa}$ enhances *neural* metrics, contributing to the realism of the edited image. Meanwhile, the inclusion of $\mathcal{L}_b$ improves our *classical* metrics, enhancing the faithfulness of the edited image. Notably, these two aspects - realism and faithfulness are orthogonal qualities in image generation and editing. The combination of both losses in LoMOE yields improved performance, achieving a balanced enhancement in both the realism and faithfulness of the edit. Detailed ablation results for varying values of $\tau$ and $T_b$, can be found in Sec. 3 of the supplementary.

## 5.4 User Study

We performed a user study using images from the *single-object* dataset to assess user preferences among images edited using the various baseline methods. We had 40 participants in the age range of 23-40. The majority of them expressed a preference for the edits generated by LoMOE over those from the other baseline methods. The results are summarized in Fig. 6, and our observations from the user preference survey are as follows:

The user study revealed that LoMOE is the most favored image editing method, with 46% of participants ranking it as their first preference and 37% as their second preference. Users expressed overall satisfaction with LoMOE's reliability, even in cases where edits weren't entirely successful. Following LoMOE, BLD and I-P2P garnered appreciation, with 25% and 13% respectively for first preference. However, BLD's failures were noted to be drastic, rendering some images unusable, while I-P2P's unintended background changes often resulted in visually appealing edits. GLIDE, DiffEdit, and SDEdit emerged as the least preferred methods, with only single-digit percentages for first preference. Dissatisfaction stemmed from GLIDE's tendency to replace subjects with poor-quality targets, and users found DiffEdit and SDEdit to be similar, with the former preserving unmasked regions of input images. Overall, LoMOE stood out as the preferred choice, while BLD and I-P2P offered viable alternatives despite their drawbacks.

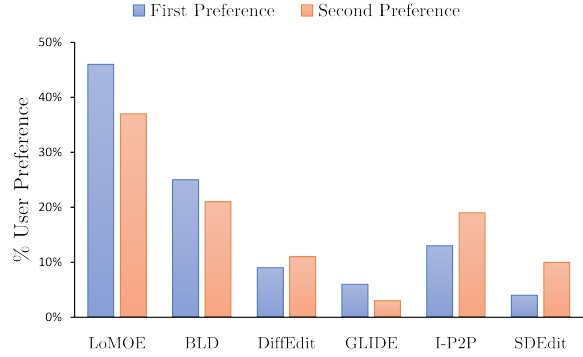

**Figure 6: User Study: The first & second preference images for users who were shown results produced by SOTA methods.**

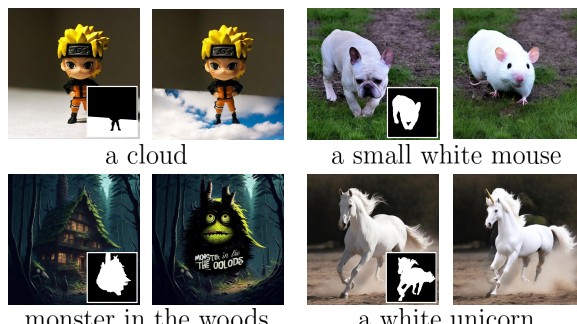

a cloud      a small white mouse

monster in the woods      a white unicorn

**Figure 7: Illustrating LoMOE's limitations, we reveal challenges in realism and its ineffectiveness to handle size or pose changes, stemming from its mask-based nature. These limitations highlight promising avenues for future research.**

## 6 LIMITATIONS

The limitations of LoMOE are illustrated in Fig. 7. These limitations are inherent to its underlying architecture, which is shared by the broader class of models it belongs to. Although LoMOE utilizes stable diffusion for generation, there are instances where, despite generating a very high fidelity edit, the quote "monster in the woods" also appears on the body (**Row 2, Col 1**, Fig. 7) due to the model interpreting the prompt as a text generation task [2]. Additionally, although the model adheres to the prompt in adding clouds to the masked region (**Row 1, Col 1**, Fig. 7), the edit is not very realistic, which can be attributed to the realism and faithfulness trade-off, as discussed in Sec. 5.1. Furthermore, similar to other mask-based generation methods, our model faces constraints in generating beyond specified regions, such as changes in pose or scale, where the input and output silhouettes of the object in question differ. This limitation is evident in the mouse and unicorn edits (**Col 2**, Fig. 7), where the model is constrained by the mask and is, therefore, unable to create a smaller mouse inside the mask or the unicorn horn outside the mask. However, it is essential to recognize that this limitation prevents unintended edits, distinguishing mask-based editing models from mask-free editing frameworks. Despite these constraints, our model demonstrates effectiveness within its scope of capabilities while maintaining precision in complex edits.

## 7 CONCLUSION

We present LoMOE, a framework designed to address a task of localized multi-object editing using diffusion models. Our approach enables (mask and prompt)-driven multi-object editing without the need for prior training, allowing diverse operations on complex scenes in a single pass, thereby having improved inference speed compared to iterative single-object editing methods. Our framework achieves high-quality reconstructions with minimal artifacts through cross-attention and background preservation losses. Further, we curate LoMOE-Bench, a benchmark dataset that provides a valuable platform for evaluating multi-object image editing frameworks. We believe that LoMOE would serve as an effective tool for artists and designers.

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
