# OpenReview forum: "LoMOE: Localized Multi-Object Editing via Multi-Diffusion"
_acmmm.org/ACMMM/2024/Conference — MM2024 Poster_

### Official Review · Reviewer_1TjC · 2024-05-19

**Rating:** 5
**Confidence:** 3

**Summary:**

This paper proposed a novel method for zero-shot localized multi-object editing, dubbed LoMOE. It utilizes foreground masks and corresponding simple text prompts that exert localized influences on the target regions resulting in high-fidelity image editing. For preserving the characteristics of the object being edited and the background, it includes a combination of cross-attention and background preservation losses. The authors also release a dataset for multi-object editing, named LoMOE-Bench. Multiple experiments are conducted to show that LoMOE achieves improvement in both image editing quality and inference speed.

**Strengths:**

1.The writing is clearly and easy to follow. The figure 2 provides a detailed depiction of the entire process of the proposed method. The authors introduce cross-attention loss  and background preservation loss  for high-quality and background preservation, which are simple and reasonable, resulting in excellent editing results.
2.The paper provides a substantial number of experiments, which are extensive and comprehensive. The authors compare the method with SOTA in both single object editing and multi-object editing tasks. They also provide a dataset dedicated to multi-object editing, which is to provide new directions for future research.

**Limitations:**

1. The cross-attention preservation loss updates each attention map of multi-diffusion process. However, previous works have shown the attention maps trend to overlap for similar objects, which could result in poor quality. Therefore, using masks in cross-attention loss may prevent such a problem. Why not use masks in the cross-attention preservation loss ?

2. The previous methods which optimize latent or attention map during the inference process usually employ the thresholds to prevent excessive interference. However, it seems that the joint optimization does not apply such thresholds. Hope the authors explain the reason for this.

3. There are still some experimental issues:
 - a.  In multi-object editing, the authors did not provide a qualitative comparison with other methods.
 - b.  Null text achieves good results in non-region editing, and it can also facilitate multi-object editing with text conditions. Hope the qualitative experiments comparing it with the proposed method can be added.
 - c. Hope to add a user study on multi-object editing.

**Suitability:**

3

---

### Official Review · Reviewer_qRuS · 2024-05-23

**Rating:** 3
**Confidence:** 3

**Summary:**

This paper proposes LoMOE, a framework for localized multi-object editing using diffusion models. LoMOE enables precise and simultaneous modifications to multiple objects within an image by utilizing foreground masks and simple text prompts. It achieves high-fidelity image edits without prior training, making it a zero-shot solution. The framework is evaluated through both single-object and multi-object editing experiments, demonstrating its effectiveness. Additionally, this paper introduces a new benchmark dataset, LoMOE-Bench, dedicated to multi-object editing. This dataset facilitates the evaluation and comparison of different image editing frameworks.

**Strengths:**

1. Compared to most methods that can only handle single-object editing, LoMOE can process multi-object editing in one go, improving editing efficiency.
2. As a zero-shot solution, LoMOE is ready to use without the need for prior training, which simplifies the usage process and lowers the application barrier.
3. The release of the new benchmark dataset LoMOE-Bench provides a standardized platform for the evaluation of multi-object image editing.
4. The popularity of LoMOE's editing results was assessed through user research, demonstrating its potential and user acceptance in practical applications.

**Limitations:**

1.	The titles of the paper are inconsistent. The title in the submitted PDF is “LoMOE: Localized Multi-Object Editing via Diffusion,” while the title in the OpenReview submission system is “LoMOE: Localized Multi-Object Editing via Multi-Diffusion.”
2.	On line 193 of the paper, the number of parentheses does not match. There are two left parentheses and one right parenthesis.
3.	Equation 8 is confusing. The function $\epsilon_\theta()$ can only accept one $x_t$ and one $c_i$ as inputs at a time.
4.	In Equation 9, the dimensions of $\bar{A}_t^r$ and $\bar{A}_t^e$ might differ because $c_0$ and $c_i$ may contain different numbers of words.
5.	The illustration in the third step (multi-object edit) of the overall framework (Figure 2) is very confusing and seems inconsistent with the method described in Section 3.3.3. Figure 2 shows that $y_{t-1}^*$ is obtained through $x_t - \nabla$. However, Equations 11, 12, and 13 in Section 3.3.3 indicate that $y_{t-1}^*$ is obtained through $\Phi{ (x_t - \nabla) }$.
6.	The notation in the paper is confusing. In Figure 2 and Equation 10, $y_t^*$ should be represented as $y_{t-1}^*$ because it is derived from $\epsilon_\theta(x_t)$.
7.	The method section of the paper is not clearly written. There is a discrepancy between the overall framework and the text in the methods section.

**Suitability:**

3

---

### Official Review · Reviewer_3bLF · 2024-05-24

**Rating:** 3
**Confidence:** 3

**Summary:**

This paper proposes a diffusion model-based framework, LoMOE, for zero-shot localized multi-object editing tasks. The proposed framework consists of three key steps: inverting the input image to obtain the latent code; applying a MultiDiffusion process for localized multi-object editing; and preserving attributes and background using cross-attention maps and a background mask. For the multi-object editing task, they curated a new dataset, LoMOE-Bench, which contains 1000 edit operations featuring images with 2 to 7 masks, each paired with corresponding text prompts. The authors compare the proposed method with existing state-of-the-art methods and show the improved effectiveness of their approach in terms of both image editing quality and inference speed.

**Strengths:**

1. The authors curate a dataset LoMOE-Bench for the multi-object editing task.
2. The paper is well-written and easy to follow.

**Limitations:**

1. The contribution of this paper is incremental to MultiDiffusion. The zero-shot localized multi-object editing is mainly achieved by MultiDiffusion, and the preservation of structures is primarily achieved via cross-attention maps following Prompt-to-prompt. A thorough discussion of the differences between the proposed method and previous works would help reviewers better understand the contribution.
2. MultiDiffusion is not chosen as one of the baseline methods in the comparison experiments. The comparison experiments in Fig.3, Fig.5, Tab.1, and Tab.2 mainly show the advantage of MultiDiffusion over the other methods.
3. As this method uses explicit masks during optimization, the proposed method might not work well when the desired target object has a different shape. Other comparison methods, such as Prompt-to-prompt, which do not use explicit masks, could handle these cases better.
4. In the first two rows in Fig. 5, the results obtained by the proposed method (last column) show the replaced objects appearing brighter and less well blended with the original background. How well this method can scale to more objects if the provided mask is larger than five? Also, the optimization time will scale when providing more masks. The optimization time is not well discussed in the implementation details.

**Suitability:**

2

---

### Official Review · Reviewer_6iAg · 2024-05-27

**Rating:** 5
**Confidence:** 3

**Summary:**

The paper proposes a novel localized editing method for generated images. The framework performs inversion as well as a multi-diffusion reconstruction process to preserve the background regions to be untouched.

**Strengths:**

- The proposed method outperforms existing methods.
- A novel benchmark is introduced to thoroughly evaluate localized image editing.

**Limitations:**

- Does the proposed method operate on real images? Are the images required to be t2i generated?
- What is the overall runtime of the proposed method?
- How are the images, prompts, etc. selected when constructing LoMOE-Bench? What are the criteria?
- Missing references on recent advances in multi-object editing [1][2].

[1] Tunanyan H, Xu D, Navasardyan S, et al. Multi-Concept T2I-Zero: Tweaking Only The Text Embeddings and Nothing Else[J]. arXiv preprint arXiv:2310.07419, 2023.
[2] Khandelwal A. Infusion: Inject and attention fusion for multi concept zero-shot text-based video editing[C]//Proceedings of the IEEE/CVF International Conference on Computer Vision. 2023: 3017-3026.

**Suitability:**

3

---

### Meta-Review · Area_Chair_i1h2 · 2024-07-01

**Recommendation:** Accept (Poster)
**Confidence:** 5

**Metareview:**

This paper was reviewed by four experts, received two week accept, one borderline accept, and one borderline reject. The paper presents LoMOE for localized multi-object editing using diffusion models. LoMOE localizes multiple objects within an image by utilizing foreground masks and edits the masked content by text prompts. The pros including building a dataset LoMOE-Bench for the multi-object editing task, well-written, good performance, etc.

The concerns raised by reviewers are mainly focus on method novelty, especially comparison with previous methods like MultiDiffusion. The AC checked the rebuttal and was convinced, but suggested the authors to include the discussion and add more comparison with object editing works like Object-level Image Editing via Referring Expressions in the final main paper. The authors are required to include the experiments and comparison with suggested references to the final version. A through proofreading/polishing is suggested to improve clarity and avoid typos.

Based on the reviews, AC found the proposed dataset LoMOE-Bench has potential benefit to the community, the AC recommend this paper for acceptance.